# Individual Survival Distributions Generated by Multi-Task Logistic Regression Yield a New Perspective on Molecular and Clinical Prognostic Factors in Gastric Adenocarcinoma

**DOI:** 10.3390/cancers16040786

**Published:** 2024-02-15

**Authors:** Daniel Skubleny, Jennifer Spratlin, Sunita Ghosh, Russell Greiner, Daniel E. Schiller, Gina R. Rayat

**Affiliations:** 1Department of Surgery, Faculty of Medicine and Dentistry, College of Health Sciences, University of Alberta, Edmonton, AB T6G 2R3, Canada; ds9@ualberta.ca (D.E.S.); grayat@ualberta.ca (G.R.R.); 2Department of Oncology, Faculty of Medicine and Dentistry, College of Health Sciences, University of Alberta, Edmonton, AB T6G 2R3, Canada; jennifer.spratlin@albertahealthservices.ca (J.S.); sunita.ghosh@ualberta.ca (S.G.); 3Department of Mathematical and Statistical Sciences, Faculty of Science, College of Natural and Applied Sciences, University of Alberta, Edmonton, AB T6G 2R3, Canada; 4Department of Computing Science, Faculty of Science, College of Natural and Applied Sciences, University of Alberta, Edmonton, AB T6G 2R3, Canada; rgreiner@ualberta.ca; 5Alberta Machine Intelligence Institute, Edmonton, AB T5J 3B1, Canada

**Keywords:** survival prediction, individual survival distributions, gastric cancer, TCGA, ACRG, molecular subtype, multi-task logistic regression

## Abstract

**Simple Summary:**

Traditional survival models estimate risk across many patients, and thus, translating survival outcomes to individual patients is a difficult task. Individual survival distributions (ISDs) provide an accurate survival curve for each individual patient. In this study, we demonstrate that ISDs established with multi-task logistic regression (MTLR) models provide accurate individual survival predictions for gastric cancer. Because MTLR is not bound by traditional assumptions, we demonstrate that the degree to which a tumour has a favourable immune system interaction has the most relevant long-term survival effects.

**Abstract:**

Recent advances in our understanding of gastric cancer biology have prompted a shift towards more personalized therapy. However, results are based on population-based survival analyses, which evaluate the average survival effects of entire treatment groups or single prognostic variables. This study uses a personalized survival modelling approach called individual survival distributions (ISDs) with the multi-task logistic regression (MTLR) model to provide novel insight into personalized survival in gastric adenocarcinoma. We performed a pooled analysis using 1043 patients from a previously characterized database annotated with molecular subtypes from the Cancer Genome Atlas, Asian Cancer Research Group, and tumour microenvironment (TME) score. The MTLR model achieved a 5-fold cross-validated concordance index of 72.1 ± 3.3%. This model found that the TME score and chemotherapy had similar survival effects over the entire study time. The TME score provided the greatest survival benefit beyond a 5-year follow-up. Stage III and Stage IV disease contributed the greatest negative effect on survival. The MTLR model weights were significantly correlated with the Cox model coefficients (Pearson coefficient = 0.86, *p* < 0.0001). We illustrate how ISDs can accurately predict the survival time for each patient, which is especially relevant in cases of molecular subtype heterogeneity. This study provides evidence that the TME score is principally associated with long-term survival in gastric adenocarcinoma. Additional external validation and investigation into the clinical utility of this ISD model in gastric cancer is an area of future research.

## 1. Introduction

Recent advances in our understanding of gastric cancer biology have prompted a shift towards personalized management. Transcriptomes or multi-omics-based molecular classification systems, such as those proposed by the Cancer Genome Atlas (TCGA) and the Asian Cancer Research Group (ACRG), have identified tumours with distinct molecular characteristics with potential implications for personalized anti-cancer therapies [1,2,3]. The identification of microsatellite instability (MSI) due to deficiency in mismatch repair proteins has enabled the use of immune checkpoint inhibitor therapy in metastatic and neoadjuvant settings [4,5,6]. The possible omission of chemotherapy in tumours with MSI is another intriguing and ongoing development [7,8,9,10,11].

Despite these advances, the outcomes of clinical trials and treatment allocation in the clinic still rely on population-based survival analysis, which evaluates the average survival effects of entire treatment groups or prognostic variables. One intriguing method to augment personalized cancer treatment is to use individual survival distributions (ISDs) [12]. ISDs provide survival estimates and visual curves for each patient based on their unique tumour and clinical characteristics. These models offer advantages over other personalized survival estimation methods, such as nomograms, because they can provide survival estimates at every future time point, and they provide an additional visual interpretation of survival probability. Nomograms function as single-time risk estimators that serve to measure the probability of whether a patient survives at, for instance, one year or three years. They do not provide any information regarding the chance of survival outside of the designated periods.

Furthermore, some ISD models, such as multi-task logistic regression, are not bound by the proportional hazards assumption, which assumes that the effect of every variable remains constant over time [13,14]. It is unlikely that every variable related to oncologic outcomes follows this assumption. For example, a chemotherapeutic treatment likely provides a proportionally greater survival effect around the time the treatment is administered, but longer-term survival may be more associated with the disease stage. Because MTLR is not bound by the proportional hazards assumption, the model can predict that patient P1 is more likely to survive 1 year than patient P2, but P2 is more likely to survive 5 years—the survival curves can cross [15]. We can also determine how the relative survival effects of a given variable change over time.

Here, we construct ISDs from a pooled set of gastric cancer patients annotated with integrated molecular classifications from the TCGA, ACRG, and tumour microenvironment (TME) score classification schemes [1,2,16]. Using MTLR, we identify a novel understanding of how clinical and molecular features vary in their survival effects over time. We also provide examples of how ISDs may be presented for individual patients and explain how survival curves can facilitate counterfactual reasoning.

## 2. Materials and Methods

### 2.1. Dataset

We used previously characterized data from a pooled integrated molecular classification dataset, which contains 2202 gastric adenocarcinoma patients from 11 publicly available datasets [17]. In this prior work, supervised machine learning models were used to produce models that classify each patient according to their TCGA, ACRG, and TME molecular subtypes. These models were used to learn the molecular subtypes of all 2202 patients. We included 1043 out of 2202 patients who possessed complete clinical data for our variables of interest, which comprised the TCGA, ACRG, and TME molecular subtype scores and age, stage, sex, Lauren classification, tumour location, and chemotherapy exposure status. Patients with missing data for these variables were excluded.

### 2.2. Models and Statistical Analysis

Individual survival distribution models for overall survival were implemented as suggested by Haider et al. [18]. The codebase for implementation in R can be found at https://github.com/haiderstats/ISDEvaluation (accessed on 16 June 2021). We tested the performance of Cox with Kalbfleisch–Prentice extensions (Cox-KP), ElasticNet Cox (CoxEN-KP), random survival forest (RSF), accelerated failure time (AFT), and multi-task logistic regression (MTLR) models for our survival prediction task. Model performance and optimization were assessed using 5-fold cross-validation (CV) [12].

The models were selected with consideration of their concordance index (C-Index), 1-calibration and D-calibration metrics [12]. The concordance index is a measure of a model’s discrimination—the proportion of pairs of subjects/patients where the prediction (i.e., model score/risk) is concordant with the true outcome [18,19]. Calibration is a measure of how well model predictions match the true observed event rate. The 1-calibration metric is simply the Hosmer–Lameshow goodness-of-fit test, which evaluates whether the predicted rate of an event is statistically similar to the true event rate at a specific time point [20]. Distribution calibration (D-calibration) assesses whether the proportion of patients predicted to experience an event is uniformly distributed in each decile using a Pearson’s χ^2^ test. This metric provides a means to determine if we should believe the predictions made by an ISD model [18]. We report the *p* value from these tests for which the null hypothesis denotes a calibrated test. If *p* < 0.05, then the model is not calibrated for that calibration metric.

The chosen time-to event models represent some common survival modelling methods but are not exhaustive. Detailed descriptions of these models have been provided in previous studies that used Cox-KP, CoxEN-KP, RSF, AFT, and MTLR models as benchmark comparisons for ISDs [12,18]. We have included brief descriptions of these models below. The Cox model is a semi-parametric model that provides a risk score. To produce an ISD, the Kalbfleisch–Prentice estimator is employed to estimate a baseline hazard function [21]. The CoxEN-KP model uses ElasticNet (EN) regularization of the negative log of the partial likelihood in an effort to improve the model fit [22,23]. Cox-KP and CoxEN-KP provide excellent discrimination (C-Index) but comparatively provide worse calibration indices [12]. The RSF is a non-parametric ensemble estimator that does not obey the proportional hazards assumption [24]. RSF is most effective in higher-dimensional datasets with low censor rates [12]. Similar to Cox, RSF provides excellent discrimination but poor calibration compared to MTLR. AFT is a parametric survival model based on the Weibull distribution and is effective in simple, low-dimensional survival prediction tasks [25]. MTLR generates individual patient survival distributions for a specific number of times based on the number of uncensored patients in a dataset [26]. An empirical study demonstrated that MTLR provided equivalent or superior discrimination compared to other survival models and provided excellent model calibration in both low- and high-dimensional datasets [12,15,18].

The survival probability and median survival time were calculated using spline functions in R. A monotonic cubic spline function using Hyman filtering of the Hermite spline method was used to generate a prediction function [27]. The median survival time was calculated using the integral (i.e., the area under the curve) of the monotonic function at a 50% survival probability. For additional details and the codebase regarding these calculations, please see https://github.com/haiderstats/ISDEvaluation (accessed on 16 June 2021).

The survival effects were assessed using the mean model weights derived from out-of-fold data (i.e., unseen data) in the 5-fold CV. The 5 most influential variables, defined as the largest mean absolute value of the MTLR weights, were selected. The survival weights as a function of time were plotted using ggplot2 version 3.4.4 as loess smooth curves, with 95% confidence intervals for the top 5 variables [28]. A forest plot was developed using the mean model weights, and the 95% confidence interval was calculated using 1000 bootstraps with replacement. We assessed the significance of each covariate using the one-sample Wilcoxon test, assuming the null hypothesis that a model weight of zero provides no survival effect. The Benjamini–Hochberg method was used to correct for multiple comparisons. Statistical significance was defined as *p* < 0.05. A Cox proportional hazards model, which was separate from the ISD Cox models, was also developed using the same patient data for the MTLR model. This Cox model was constructed without cross-validation and instead used all 1043 patients to generate the Cox coefficients. The similarity between the Cox regression coefficients and the MTLR model weights was assessed using Pearson’s correlation.

## 3. Results

We constructed ISDs to expand on the utility of our integrated molecular classification models using continuous model probability scores. Using 1043 patients with available clinicopathologic characteristics (Table 1), we evaluated the performance of several ISD models for our prediction task (Table 2). Multi-task logistic regression provided a superior calibrated model with a nearly identical C-Index compared to CoxEN-KP (MTLR C-Index = 72.1 ± 3.3% versus CoxKP-EN = 72.2 ± 2.9%). The MTLR model was D-calibrated and 1-calibrated for all bins except the 50% percentile (Table 2).

In contrast to Cox, MTLR is not bound by the proportional hazards assumption. We evaluated whether MTLR could provide unique insight into survival effects by modelling the weights derived from out-of-fold patients in the 5-fold CV. In Figure 1A, the loess smooths and their 95% confidence intervals illustrate the relationship of the top five most influential covariates to survival over the entire time course of our model. As opposed to presenting a consistent risk over time, this approach suggests that certain covariates present more prominent effects at different times from disease presentation. For example, early beneficial effects are observed for chemotherapy, which taper off after 24 months. Stage IV disease exerted a constant negative survival effect over 10 years, but Stage III disease did not provide increased death until after one year. Notably, a greater TME score was most associated with long-term survival beyond 5 years and closely mirrored the temporal effects of chemotherapy.

To enhance familiarity with ISD models we assessed whether the estimates from the MTLR models are similar to the “gold standard” Cox regression coefficients. In Figure 1B, negative weights derived from the MTLR model correspond to improved survival. Here, we present the mean weights for all predicted time points with their respective 95% confidence intervals. Stages III and IV provided the greatest effect size on survival (mean weight of 0.027 (95% CI 0.022, 0.033); *p* < 0.001, and mean weight of 0.049 (95% CI 0.046, 0.053); *p* < 0.001, respectively). Survival was also significantly decreased by age, Stage II, and increasing epithelial-to-mesenchymal transition (EMT) and chromosomal instability (CIN) scores. The most beneficial survival effects were observed for chemotherapy and increasing high TME scores (mean weight of −0.021 (95% CI −0.027, −0.016); *p* < 0.001, and mean weight of −0.015 (95% CI −0.020, −0.010); *p* < 0.001, respectively). Microsatellite instability, as defined by our TCGA classifier, significantly improved survival, whereas ACRG MSI was not significant. With this appealing presentation, MTLR can provide familiar interpretations of survival effects relative to traditional Cox models.

Next, we compared the similarity of the coefficients of the Cox proportional hazards model to the mean weights derived from the MTLR model. We show that there are strong correlations (Pearson coefficient = 0.86, *p* < 0.0001) among the model effects for the variables of interest in Figure 1C. Thus, an ISD using MTLR provides similar population-based interpretations as Cox models when averaged over all predicted time points but also provides advantages over Cox, including better calibration and the absence of the proportional hazards assumption.

We evaluated the ability of our MTLR model to provide insights into personalized medicine for gastric cancer. Three scenarios are presented in Figure 2A. Scenario a (i.e., blue curves) presents the effects of chemotherapy in two Stage II males in their sixties with similar TME scores. Here, the patient who received chemotherapy had a 13.8% and 18.7% greater probability of survival at 24 and 48 months, respectively. Scenario b (i.e., red curves) illustrates the relationship between the TME score and chemotherapy in Stage III chromosomal instability (CIN) tumours. We observed that the survival curves for a patient who did not receive chemotherapy are nearly identical to one who received chemotherapy with a low TME tumour. MTLR provided additional insight into the individual survival effects of other molecular subtypes. In Scenario c (i.e., purple curves), we found that a high epithelial-to-mesenchymal (EMT) score profoundly affected the overall survival in comparable Stage IV, chemo-naïve, and low TME tumours (median survival with high EMT = 9.4 months versus low EMT = 14.5 months).

We investigated potential counterfactual applications of ISD models to facilitate the communication of personalized medicine. In Figure 2B, we demonstrate the survival benefit, as interpreted by our MTLR model, of administering chemotherapy to a 67-year-old female with a Stage IV, high-TME-score gastric cancer. Although additional research is required, counterfactual scenarios could be presented to patients as a visual interpretation of otherwise foreign and abstract statistical estimates of treatment benefits/harms.

## 4. Discussion

This study presents a survival modelling perspective called individual survival distributions with the MTLR model to provide novel insight into personalized survival predictions of gastric adenocarcinoma. Prior research has demonstrated that ISDs provide a valuable alternative to popular personalized survival modelling strategies, such as nomograms [12,15]. Using MTLR, we identified that the TME score and chemotherapy provide non-proportional but similar survival effects over time. In our analysis, the TME score provided the greatest beneficial survival effect beyond 5 years. Conversely, the disease stage was the most prominent prognostic factor contributing to poor patient survival. Of note, prominent molecular classification subtypes derived from the TCGA and ACRG cohorts were not among the top 5 most influential variables for survival.

Personalized survival interpretation using ISDs provides an intuitive and visually appealing method to investigate and present survival effects. In addition to illustrating individual survival curves, we also demonstrate that the MTLR model can provide population-based interpretations of survival effects similar to Cox proportional hazards when the model weights are averaged over the entire time modelled. In this study, we quantitatively demonstrated the statistically significant similarity between the Cox model coefficients and the MTLR model weights. Our intention for this comparison is to convey a sense of familiarity with the MTLR model for researchers who may typically use Cox regression. The ubiquitous Cox model was effectively designed to produce an effective discriminatory tool (i.e., C-Index). However, it suffers from poor calibration, which motivated us to consider more general models, such as MTLR. In this study, we empirically confirm that MTLR provides nearly equivalent model discrimination but also better calibration. Thus, we propose that ISDs using MTLR is a valuable tool for survival analysis and may be used in addition to or instead of traditional Cox models. It is important to note that the interpretation of the survival effects of MTLR variables is relative to the variables included in that specific model. This interpretation also applies to Cox models and any other survival model generated using observational data.

Specific to gastric cancer, our MTLR model approach builds on previous research that shows that the tumour immune microenvironment is a significant prognostic variable [2,16,17]. The TME score was developed using an in silico analysis of the ACRG microarray-based molecular classification of gastric cancer. The TME score is characterized by immune activation, the response to the virus, and the interferon-gamma response, as well as enrichment in chemokines, such as *CXCL10*, *CXCL9*, and *CCL4* [16]. It was also demonstrated to be a predictive biomarker for the immune checkpoint inhibitor response in advanced melanoma and metastatic urothelial cancers. The TME score is strongly associated with EBV-type and MSI gastric cancer, which have been demonstrated to possess immune-rich microenvironments [2,17]. In an era of immunotherapy, a personalized survival model informed by the TME score could provide a valuable tool in allocating immune checkpoint blockade therapy.

Our results provide insight into how gastric cancer survival may be optimized. For example, age, Stage II, and Stage III represent the majority of the top five most influential variables for survival. These variables are unmodifiable at the time of diagnosis for a given patient. The only method to improve these prognostic factors would be to identify disease at an earlier stage and age. Although this strategy is admirable, universal screening in low-incidence populations, such as those in Western nations, is not feasible. However, first-generation immigrants from high-incidence countries comprise one screening population that could be targeted in low-incidence nations [29,30,31]. In contrast to age and stage, the other top five prognostic factors, namely chemotherapy and TME score, are potentially modifiable factors. The choice of an individual patient to forgo chemotherapy is a significant factor for survival. Thus, conveying the importance of chemotherapy using ISDs is one method that may improve adherence to chemotherapy treatment. Of course, the use of chemotherapy must be balanced with the patient’s performance status and goals of care.

The TME score is the most intriguing modifiable risk factor. The efficacy of adjuvant and neoadjuvant chemotherapy/chemoradiotherapy, as well as checkpoint-inhibition therapy, has been demonstrated to be associated with the tumour immune microenvironment [32,33,34,35]. Thus, augmenting the tumour immune microenvironment is an intriguing therapeutic strategy [36,37]. Two potential methods to achieve this goal include cancer vaccines or chemokine/cytokine therapy. In a landmark Phase 1B trial, Rojas et al. demonstrated that personalized mRNA BioNTech vaccines containing unique tumour neoantigens significantly augmented immune microenvironments and survival in pancreatic adenocarcinoma patients [38]. Relevant chemokine/cytokine-directed strategies include combining IL-12 plasmid therapy with immune checkpoint inhibition, or a CXCR4 antagonist and pembrolizumab with chemotherapy, among others [39,40].

There are several limitations of our observational study. The population consisted mainly of patients from Asian countries. Furthermore, there is potential for selection bias, as only patients with publicly available whole-transcriptome data were included. Additionally, the chemotherapy regimens were heterogeneous and only consisted of adjuvant chemotherapy. Thus, the generalizability of our findings must be validated in the context of neoadjuvant chemotherapy, which is the predominate treatment regimen in Western countries [41].

There is currently no data evaluating the utility of ISDs in improving patient outcomes, physician decision making, or patient education. Indeed, despite the exciting prospects of ISDs, their value must be shown in clinical care. Future studies should include patient-reported outcomes and qualitative analyses of the perception of ISDs versus other prognostic tools, such as physician-directed discussions or computer-based tools, such as nomograms. We hypothesize that the visual presentation of survival in an individualized and appealing graphical format may provide an improved perspective on disease severity, decrease patient uncertainty regarding their diagnosis, and enhance confidence in patient-centered decision making. A similar sentiment regarding nomograms is well-characterized. Regardless of model accuracy or calibration performance, personalized survival models are most valuable if they ultimately improve patient and physician satisfaction and survival outcomes [42].

To facilitate the use of ISD models and the MTLR model, we provide links and citations in Section 2 of this manuscript to the R codebase used to implement ISD models, the MTLR R package, and the Python package Survival EVAL [13,43].

## 5. Conclusions

Individual survival distributions using the MTLR model provide novel insight into gastric cancer prognostic factors in the context of a pooled analysis of publicly available data annotated with integrated molecular subtype classification. Additional external validation and investigation into the clinical utility of this ISD model is an area of future research.

## Figures and Tables

**Figure 1 cancers-16-00786-f001:**
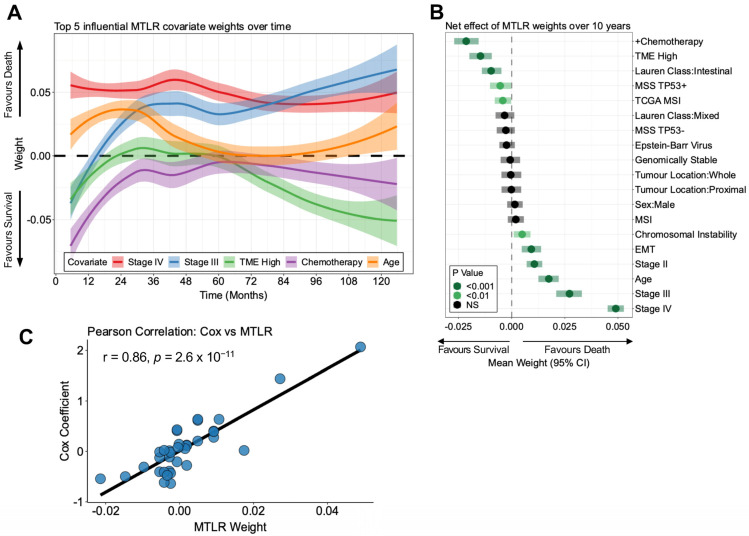
Individual survival distributions using multitask logistic regression. (**A**) Loess smooth curves for the top 5 most influential covariate MTLR weights generated using 5-fold cross-validation. Shaded bands represent 95% confidence intervals. (**B**) Mean MTLR weights averaged over all time points for each covariate in our model. The point represents the mean, and the semi-transparent box represents the 95% confidence interval estimate from 1000 bootstraps. Weights less than zero favour survival. A one-sample *t*-test was performed to assess if a covariate was significantly greater than zero, and *p* values were corrected using Benjamini–Hochberg. The *p*-value significance is denoted in the plot legend. (**C**) Scatter plot of Cox proportional hazards model coefficients versus MTLR weight for each variable. The Pearson correlation coefficient and *p*-value are denoted in the plot. Acronyms: MSS TP53+ = microsatellite stable tp53 positive; MSS TP53− = microsatellite stable tp53 negative; EMT = epithelial-to-mesenchymal transition.

**Figure 2 cancers-16-00786-f002:**
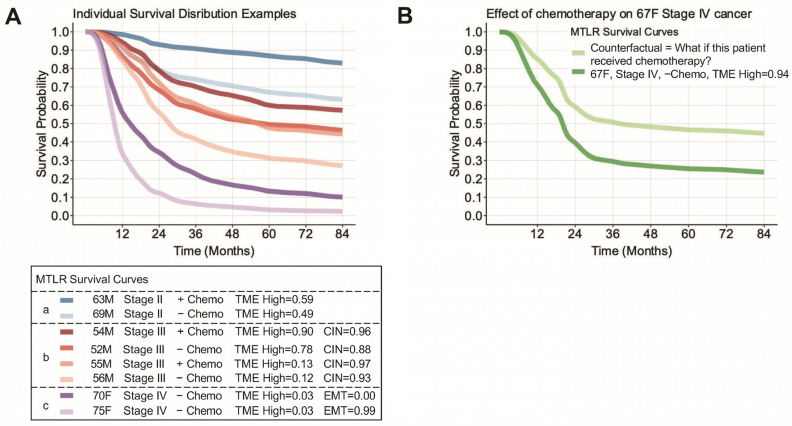
Generation of personalized survival curves using ISDs. (**A**) Individual survival curves for 8 patients using a learned MTLR model. The *x*-axis represents time in months, and the *y*-axis represents survival probability. The patient characteristics for each colour are shown in the plot legend below. Scenario a (i.e., blue curves) presents the effects of chemotherapy in two Stage II males in their sixties with similar TME scores. Scenario b (i.e., red curves) illustrates the relationship between the TME score and chemo-therapy in Stage III chromosomal instability (CIN) tumours. Scenario c (i.e., purple curves), Stage IV, chemo-naïve, and low TME tumours. (**B**) Example of a counterfactual scenario illustrating the predicted effect of chemotherapy for a given patient.

**Table 1 cancers-16-00786-t001:** Patient demographics.

Characteristic	n/N (Missing %)	N = 1043 ^1^
**Age**	1043/1043 (0%)	59 (49, 67)
**Stage**	1043/1043 (0%)	
I		170 (16%)
II		330 (32%)
III		339 (33%)
IV		204 (20%)
**Sex**	1043/1043 (0%)	
Female		359 (34%)
Male		684 (66%)
**TCGA Subtype**	1043/1043 (0%)	
Chromosomal Instability		824 (79%)
Epstein–Barr Virus Type		43 (4.1%)
Genomically Stable		66 (6.3%)
Microsatellite Instability		110 (11%)
**ACRG Subtype**	1043/1043 (0%)	
Epithelial-to-Mesenchymal Transition		118 (11%)
Microsatellite Instability		162 (16%)
Microsatellite Stable TP53 Negative		412 (40%)
Microsatellite Stable TP53 Positive		351 (34%)
**TME Subtype**	1043/1043 (0%)	
High		478 (46%)
Low		565 (54%)
**Lauren Classification**	1043/1043 (0%)	
Diffuse		495 (47%)
Intestinal		504 (48%)
Mixed		44 (4.2%)
**Tumour Location**	1043/1043 (0%)	
Distal		537 (51%)
Proximal		482 (46%)
Whole		24 (2.3%)
**Treatment**	1043/1043 (0%)	
No		299 (29%)
Yes		744 (71%)
**Study**	1043/1043 (0%)	
ACRG		219 (21%)
Kosin		98 (9.4%)
KUGH		82 (7.9%)
Samsung		432 (41%)
TCGA		151 (14%)
Yonsei MDACC		61 (5.8%)

^1^ Median (IQR); n (%); MDACC = MD Anderson Cancer Center; KUGH = Korea University Guro Hospital.

**Table 2 cancers-16-00786-t002:** Individual survival distribution results.

	Model				
Metric	AFT	CoxKP	CoxKPEN	MTLR	RSF
Concordance ^1^	0.720 ± 0.028	0.721 ± 0.028	0.722 ± 0.029	0.721 ± 0.033	0.699 ± 0.048
D-Calibration ^2^	0.425	0.993	0.994	0.980	0.866
1-Calibration 10th ^2^	0.147	0.447	0.898	0.086	0.354
1-Calibration 25th ^2^	0.024	0.470	0.477	0.506	0.026
1-Calibration 50th ^2^	0.000	0.011	0.027	0.042	0.447
1-Calibration 75th ^2^	0.000	0.002	0.006	0.243	0.655
1-Calibration 90th ^2^	0.000	0.009	0.027	0.132	0.050
Integrated Brier ^1^	0.176 ± 0.015	0.169 ± 0.011	0.169 ± 0.011	0.178 ± 0.015	0.178 ± 0.021

^1^ Mean ± standard deviation; ^2^ *p*-value.

## Data Availability

All data used to perform the analysis in this study are available publicly at https://github.com/skubleny/ISD_GastricCancer/tree/main (accessed on 25 November 2023).

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
