# Peer review of "Individual Survival Distributions Generated by Multi-Task Logistic Regression Yield a New Perspective on Molecular and Clinical Prognostic Factors in Gastric Adenocarcinoma"

_cancers, 2024, doi:10.3390/cancers16040786_

Round 1

Reviewer 1 Report

Comments and Suggestions for Authors

1. For the dataset use any inclusion and exclusion criteria.

2.  Statistical significance level should be mentioned in the method section as P<0.05.

3.  Please provide more elaborate suggestions for future studies and explain in detail how those studies are going to improve the patient well being. 

Author Response

Thanks to the reviewer for their time and dedication. Please see our responses below. 

Comments and Suggestions for Authors

  1. For the dataset use any inclusion and exclusion criteria.

Thank you for this suggestion. We have expanded and provided inclusion and exclusion criteria listed in the first paragraph of Materials and Methods.

  1. Statistical significance level should be mentioned in the method section as P<0.05. 

The definition of statistical significance is now defined in the methods.

  1. Please provide more elaborate suggestions for future studies and explain in detail how those studies are going to improve the patient well being. 

We have provided additional detail into how and why future studies on individual survival distribution models should be completed within the discussion section. We also include suggestions for outcome measures for these proposed studies.

_________

We hope that these responses and the corresponding revisions in the text are satisfactory.

Reviewer 2 Report

Comments and Suggestions for Authors

The article showed the results of using multi-task logistic regression to generate ISD fro gastric adenocarcinoma.  This article was well written and informative.  

Author Response

Thank you to the reviewer for their time in reviewing this paper.

Reviewer 3 Report

Comments and Suggestions for Authors

This study introduces Individual Survival Distributions (ISDs) generated by multi-task logistic regression (MTLR) models as a personalized survival modeling approach in gastric adenocarcinoma. Some drawbacks exist:

1. The study emphasizes that MTLR is not bound by proportional hazard assumptions, yet it compares the results with Cox proportional hazards model coefficients. The potential implications of this comparison and the underlying assumptions need to be thoroughly discussed, as it introduces complexities in interpreting the observed correlations between MTLR model weights and Cox model coefficients.

2. please clearly specify the characteristics of the 1,043 patients included in the analysis, emphasizing any selection criteria and potential biases.

3. In the "Models and Statistical Analysis" section, briefly explain the rationale for choosing specific survival models (Cox-KP, CoxEN-KP, RSF, AFT, MTLR) and their advantages and drawbacks.

4. Please clarify the significance of the Concordance Index (C-Index), 1-calibration, and D-calibration metrics in assessing model performance.

5. Please emphasize the clinical relevance of the findings, especially regarding the identified prognostic factors (TME score, chemotherapy, disease stage).

6. Considering the potential clinical implications of the author's findings, I strongly encourage the development of an R package for the Individual Survival Distributions (ISD) model. This would enhance the accessibility of the methodology.

Author Response

We found this review very helpful in improving our manuscript. Thank you for your time and dedication. Our responses follow. 

1. The study emphasizes that MTLR is not bound by proportional hazard assumptions, yet it compares the results with Cox proportional hazards model coefficients. The potential implications of this comparison and the underlying assumptions need to be thoroughly discussed, as it introduces complexities in interpreting the observed correlations between MTLR model weights and Cox model coefficients.

Thank you for this criticism. We have addressed these concerns throughout the manuscript and hope that they satisfy the reviewer. In the discussion we state that “our intention of this comparison is to convey a sense of familiarity with the MTLR model for researchers, who may typically use Cox regression.” Novel models, especially those with machine learning implications, may be ominous for researchers to apply or may be misinterpreted. Given the significant advancements in medical outcomes driven partially by evidence-based medicine using Cox models it is reassuring that MTLR provides similar population-based interpretations when the entire time period of the model is considered. There are no assumptions in our comparison. We are simply comparing the interpretations of a familiar and ubiquitous time-to-event model (Cox) with another time-to-event model (MTLR).

The main philosophical difference between MTLR and Cox is that there is no proportional hazards assumption. Thus the interpretation of a survival effect in a Cox model is the same from 0-12 months as it is from 36-72 months. From the perspective of MTLR, model variables may exert different effects from 0-12 months as they do from 36-72 months. Our correlation analysis demonstrates that when evaluated apples to apples (i.e. across the entire time-period modeled – the average model weights are comparable to the Cox coefficients. To finalize this example a researcher may be able to demonstrate that variable “Y” is the main driver of death over a 72 months span using both Cox and MTLR models. The difference is that a researcher using MTLR can interpret that the main driver of death from 0-12 months is actually variable “X” (note: Cox still provides the proportional effect saying that “Y” is the most important from 0-12 months, 12-24 months and so on), and that variable may be modifiable or identify a more nuanced view of a given disease process.

Performance represents the second difference between MTLR and Cox and it is evaluated empirically in this study. In the paper we illustrate the potential advantages of the MTLR model empirically by comparing the model head-to-head with ISD extensions of the Cox model such as Cox-KP and Cox-EN. The ubiquitous Cox was effectively designed to produce an effective discriminatory tool – C-index.  However, it is not calibrated, which motivated to consider more general models, such as MTLR. 

2. please clearly specify the characteristics of the 1,043 patients included in the analysis, emphasizing any selection criteria and potential biases.

The inclusion and exclusion criteria has been revised in the first paragraph of the methods section. A new limitations paragraph has been placed in the discussion discussing bias related to the patients selected in this study.

The characteristics for the patients in this study are outlined in Table 1.

3. In the "Models and Statistical Analysis" section, briefly explain the rationale for choosing specific survival models (Cox-KP, CoxEN-KP, RSF, AFT, MTLR) and their advantages and drawbacks.

Thank you for this suggestion. We have provided a rationale and overview of the included models in the methods section.

4. Please clarify the significance of the Concordance Index (C-Index), 1-calibration, and D-calibration metrics in assessing model performance.

Thank you for this suggestion. We have provided a rationale and overview of the included performance metrics in the methods section.

5. Please emphasize the clinical relevance of the findings, especially regarding the identified prognostic factors (TME score, chemotherapy, disease stage).

We have added additional emphasis on the clinical context/relevance of the findings in addition to those stated in paragraph 1 of the introduction and paragraph 1, 4 and 5 of the discussion.

In addition, we describe the imitations of our findings in the context of chemotherapy given that only adjuvant chemotherapy is included in this study.

6. Considering the potential clinical implications of the author's findings, I strongly encourage the development of an R package for the Individual Survival Distributions (ISD) model. This would enhance the accessibility of the methodology.

We agree that dissemination of these useful models and the ISD approach is important. In the first sentence of Methods section 2.2 we provide a link to Humza Haider’s github page that provides complete scripts to implement ISD models in R. The MTLR model itself is implemented in R as a package called MTLR that is available on CRAN. This package is citation #13.

Furthermore, we have added additional references to a python package called SurvivalEVAL: A Comprehensive Open-Source Python Package for Evaluating Individual Survival Distributions

We had added a code/implementation sentence in the discussion as well to bring further awareness to the availability of these models. 

Reviewer 4 Report

Comments and Suggestions for Authors

The inclusion of tumor microenvironment scores in survival prediction models is a necessary discussion, and individual survival prediction is also very novel.But the quality and reliability of this manuscript did not meet our requirements, so I had to make a decision to reject it for the following reasons:
1.The level of language expression in this manuscript does not meet the standards for publication, and there are many sentences in the manuscript that express unclear meanings, such as line 17-18, and the title needs to be concise to better express the theme of the manuscript.
2.The article as a whole is shorter, which is not the main problem. The key point is that the authors did not explain the methods and results clearly, and the authors cited a literature in the methods [17], which I can't find it on pubmed. If the article is not published, the author cannot cite it and needs to describe in detail the methodology used in this paper. Secondly, the results were unclear, and there was no one major outcome. The breadth and depth of the discussion needed to be increased, and the number of references needed to be significantly increased.
3.Overall, the work lacks innovation and application, and the author needs to think deeply about the topic and make the article more substantial.

Comments on the Quality of English Language

Extensive editing of English language required.

Author Response

Thank you for your time and dedication for this review. Our responses are listed below. 

1.The level of language expression in this manuscript does not meet the standards for publication, and there are many sentences in the manuscript that express unclear meanings, such as line 17-18, and the title needs to be concise to better express the theme of the manuscript.

Thank you for expressing this concern. We have read through the manuscript and made changes to enhance readability.

2.The article as a whole is shorter, which is not the main problem. The key point is that the authors did not explain the methods and results clearly, and the authors cited a literature in the methods [17], which I can't find it on pubmed. If the article is not published, the author cannot cite it and needs to describe in detail the methodology used in this paper. Secondly, the results were unclear, and there was no one major outcome. The breadth and depth of the discussion needed to be increased, and the number of references needed to be significantly increased.

This is a valid concern. We have provided a copy of this manuscript to the editor for review (if not reviewed we can provide another copy to the editor). The manuscript in question is currently under revision Clinical Cancer Research. We have received the first rounds of revisions requests and anticipate that the article will be successfully published. We will defer to the journal in managing this concern.

We also hope that the reviewer approves of the additions made to the Methods and Discussion sections. These contributions should serve to enhance the detail and length of the manuscript. 

3.Overall, the work lacks innovation and application, and the author needs to think deeply about the topic and make the article more substantial.

We acknowledge the reviewer's opinion. We respectfully offer the following rebuttal. 

We have made several edits to the manuscript that we hope will satisfy the reviewer’s concern. We are thankful to the peer-review process in enhancing the quality of this manuscript.

A coherent characterization of individual survival distributions is a recent innovation and provides numerous advantages over other personalized survival applications such as nomograms. MTLR models provide a unique perspective for time-to-event analysis beyond that of the commonly used Cox/Kaplan-Meier models. This paper uses the only dataset, to our knowledge, that integrates three relevant and well-characterized molecular classification systems in gastric cancer in a population of over 1000 patients. In regard to the question of innovation, we argue that the application of a relatively novel survival model to a patient population not previously analyzed using the MTLR model is inherently innovative.

Reviewer 5 Report

Comments and Suggestions for Authors

Dear Authors,

Your article, "Individual Survival Distributions generated by multi-task logistic regression yield a new perspective on personalized survival effects of molecular and clinical prognostic factors in gastric adenocarcinoma," presents a novel approach in predicting individual survival in gastric cancer using multi-task logistic regression (MTLR) models. While the study, involving 1043 patients from a previously characterized database, highlights the importance of tumor-immune system interactions, there are several areas that require further clarification and validation:

1.     The paper claims to offer novel insights but does not thoroughly address the statistical assumptions and potential biases inherent in the MTLR model. A detailed discussion on how these biases might affect the interpretation of results is necessary.

2.     There is a lack of external validation with independent datasets, particularly from different types of cancer, which is vital for confirming the model's applicability and accuracy across diverse patient cohorts. It would be beneficial to test the model's effectiveness in other similar cancer types to gauge its generalizability.

3.     The manuscript does not provide a potential biological explanation for the association between the Tumor Microenvironment (TME) score and long-term survival in gastric cancer. Insights into aspects like stemness, immune evasion of cells would enhance the understanding of this relationship.

Further validation and elaboration on these points are essential for the manuscript's advancement.

Best regards.

Comments on the Quality of English Language

Minor editing of English language required

Author Response

Thank you for your time and dedication to this review. 

  1. The paper claims to offer novel insights but does not thoroughly address the statistical assumptions and potential biases inherent in the MTLR model. A detailed discussion on how these biases might affect the interpretation of results is necessary.

Thank you to the reviewer for their valuable comment.

“All models are wrong, but some are useful” (Box, G. E. Science and statistics. Journal of the American Statistical Association, 71(356):791–799, 1976.)

One must empirically test whether a model is able to perform adequately with their given dataset and question asked. In this paper we evaluate the performance of 5 separate ISD models and empirically establish the utility of the MTLR model. Here we determine that there is essentially equivalent discrimination of the MTLR model compared to Cox but there is superior calibration of the MTLR model. Thus, the MTLR model is the most desirable model for this prediction task. Furthermore, given the lack of a proportional hazards assumption more nuanced interpretations are available.

To address the reviewers concerns we have added additional information regarding the pros and cons of the included survival models in the methods. Of note, in prior works, the MTLR model has been demonstrated to provide excellent performance in both simple and complex survival tasks (see Haider et al. as cited in the manscript)

Another way to view the MTLR model is to consider the difference between a Neural Network versus a linear model. Just like neural nets often work better than simple linear models (as NN are more expressive), so too does a more expressive survival model like MTLR work better than Cox, as MTLR does not require proportional hazards and can model non-linear dependency of variables.

  1.     There is a lack of external validation with independent datasets, particularly from different types of cancer, which is vital for confirming the model's applicability and accuracy across diverse patient cohorts. It would be beneficial to test the model's effectiveness in other similar cancer types to gauge its generalizability.

We appreciate the concern of proper validation expressed by the reviewer.

We argue that adequate and proper cross-validation was performed. A common issue with cross-validation in other studies is when researchers “cheat” and allow data to contaminate model interpretations/performance. Proper validation must occur with completely unseen data. Here, we performed 5-fold cross validation and tested the results on completely unseen data across 5-folds.

We also point out and have added further emphasis that we base our interpretation of our model variables on out-of-fold data in the second paragraph of the results section. Thus, we believe we provide a stringent interpretation of our results because we use the out of fold results as opposed to the traditional techniques that incorporate all data for interpretation models. For example, database research typically  uses the entire data set to create a Cox model for interpretation. Based on this model a researcher claims a particular variable is an important prognostic factor. However, using all the data makes the model prone to overfitting. Here our interpretations are based on out-of-fold data and thus more generalizable conclusions may be made.

In regard to testing this model on separate cancer types we respectfully disagree on the utility of this claim. A machine learning model’s performance is contingent on the training data. This model was not generated for the intent of evaluating across different cancer types. Extension of this model to other cancers would be inherently wrong. Much like the same way a facial recognition model is used to train faces you would not use it to recognize text. 

If the reviewer is interested in determining whether MTLR may be useful across a variety of cancers we have addressed this question in another paper (https://aacrjournals.org/clincancerres/article/29/19/3924/729105/Learning-Individual-Survival-Models-from-PanCancer). Using the PanCancer Atlas we empirically determine that building MTLR survival models on individual cancers is more accurate than a general model across multiple cancers.

  1.     The manuscript does not provide a potential biological explanation for the association between the Tumor Microenvironment (TME) score and long-term survival in gastric cancer. Insights into aspects like stemness, immune evasion of cells would enhance the understanding of this relationship.

Thank you for recommending this addition. In combination with this suggestion and those of other reviewers we have provided more in-depth discussion regarding tumour immune microenvironments, their importance and therapeutic strategies to enhance the immune microenvironment. The ideas suggested in this paper require ongoing research given that it is based on observational data.

Round 2

Reviewer 5 Report

Comments and Suggestions for Authors

I strongly disagree with the author's response on 2nd point. Cross-validation and validation with independent datasets are different and serves different purpose.

However, authors has responded to all the comments and I do not wish to delay the publication.

Best.

Comments on the Quality of English Language

Minor editing of English language required.